# LANGUAGE DECISION TRANSFORMERS WITH EXPONENTIAL TILT FOR INTERACTIVE TEXT ENVIRONMENTS

## ABSTRACT

Text-based game environments are challenging because agents must deal with long sequences of text, execute compositional actions using text and learn from sparse rewards. We address these challenges by proposing Language Decision Transformers (LDTs), a framework that is based on transformer language models and decision transformers (DTs). Our LDTs extend DTs with 3 components: (1) exponential tilt to guide the agent towards high obtainable goals, (2) novel goal conditioning methods yielding better results than the traditional return-to-go (sum of all future rewards), and (3) a model of future observations that improves agent performance. LDTs are the first to address offline RL with DTs on these challenging games. Our experiments show that LDTs achieve the highest scores among many different types of agents on some of the most challenging Jericho games, such as *Enchanter*.

## 1 INTRODUCTION

People spend a significant fraction of their lives performing activities closely linked with natural languages, such as having conversations, writing e-mails, filling out forms, reading and writing documents, and so on. Recently, the excitement around the use of Large Language Models (LLMs) for dialogue has brought the setting of interactive dialogue into the spotlight. Interactive text-based games allow one to explore and test interactive agents, alternative neural architectures, and techniques. However, text environments remain challenging for existing Reinforcement Learning (RL) agents since the action space is vast due to the compositional nature of language, making exploration difficult. Fortunately, language has the advantage that knowledge can often be reused across environments, such as the fact that fire *burns* or that doors *open*. To solve real-world text-based tasks and play rich text-based games

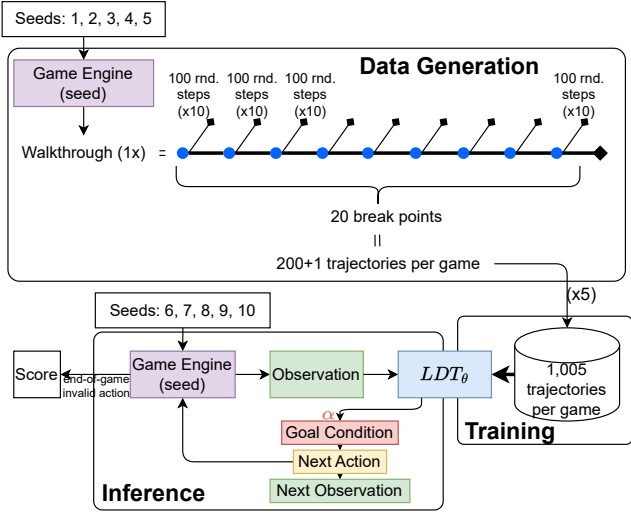

Figure 1: Overview of our approach: Noisy trajectories are generated from a high quality game walkthrough by taking 100 random steps at each 5% of the trajectory. The collection of trajectories on multiple games is used to train our LDT model offline to predict a goal condition, next action, and next observation. The LDT is then evaluated in each game environment, initialized with 5 random seeds.

well, RL agents can also benefit from the knowledge about the human world acquired from large offline data sources by leveraging pre-trained LLMs.

In real-world settings, the low-performing behavior exhibited by online RL agents during learning makes them impractical to use with humans in the loop. This situation arises in many other contexts

(Levine et al., 2020) and has motivated a lot of research on offline RL. Offline RL methods have a long history, but more recently, several approaches have been proposed that focused on using powerful transformer-based sequence models, including Trajectory Transformers (TTs) (Janner et al., 2021), and Decision Transformers (DTs) (Chen et al., 2021). However, these approaches are formulated and examined within continuous control robotics problems. Unlike the methods above, our approach is designed to handle the complexity and richness of human language by leveraging pre-trained LLMs.

Motivated by the analogy of text-games to intelligent text assistants helping people with various tasks, we assume that a few expensive expert demonstrations are available for learning. As such, we use the Jericho text games (Hausknecht et al., 2020), which provide a single golden path trajectory per game. To create a large and diverse dataset, we then generate trajectories with perturbations from that golden path as described in Section 4 and depicted in Figure 1. The complexity and richness of Jericho games make them a reasonable proxy for the kind of data one might obtain in real-world assistive agent settings.

In this work, we use a pre-trained Transformer language model that we fine-tune on offline game trajectories to predict in order: (i) a numerical trajectory quality measure used to condition the generation of the next actions (termed "*goal condition*"), (ii) next actions and (iii) future observations. To sample high-quality trajectories from our model, we convert distributions over discrete token representations of goal conditions into continuous ones, allowing us to maximize them through an exponential tilting technique. In addition, we compare different definitions of trajectory quality measures, and introduce an auxiliary loss to predict future observations. We will refer to trajectory quality measures as "*goal conditions*" in the rest of this paper and our approach as Language Decision Transformers (LDTs) with exponential tilt. Our approach is visualized in Figure 2. See Table 1 for a comparison of how our formulation for density estimation and decision-making is situated with respect to prior frameworks. We also note that none of these previous frameworks have been applied to text-based action spaces, so none have leveraged pre-trained LLMs as in our framework.

To conclude, our contributions can be summarized as follows: (1) Our work is the first to address the challenging Jericho text-based games in an offline return conditioned sequence learning setup, wherein we train models on noisy walkthrough trajectories from multiple games simultaneously. (2) We improve agent behavior with fewer assumptions by letting the model predict goal conditions in a manner where no knowledge of the maximum score is needed through our use of an exponential tilting technique. (Section 5.1). (3) We explore and empirically compare 3 novel definitions of goal conditioning that perform better than the return-to-go perspective of Decision Transformers (DTs). (Section 5.2). (4) We propose a novel auxiliary loss to train DTs that draws parallels to model-based RL and empirically shows better performance compared to the traditional model-free loss of DTs (Section 5.3). We test our proposed solutions on 33 different, realistic and complex text environments and show that LDTs performs 10% better than previous baselines on the hardest environments, and up to 30% better on average across all environments.

## 2 METHODOLOGY

### 2.1 PROBLEM SETUP

Text-based games can be formulated as partially observable Markov decision processes (POMDP) described by $(S, T, A, O, R, \gamma)$. The current game state $s_t \in S$ is partially observable in $o_t \in O$ which is often a text description of the current scene (inventory, location, items). The agent can take an action $a_t \in A$ to interact with the environment and causes a state change based on a transition function $T(s_t, a_t)$ leading to a new state $s_{t+1} \in S$. Some games are stochastic in that the same action for the same state can lead to different states. Once the agent transitions to the new state, a reward $r_t$ is given by an unknown reward function $R(s_t, a_t)$ that the game designers defined. The reward can either be positive, negative, or neutral.

**Offline Reinforcement Learning.** The goal of the agent is to learn a policy $\pi(a_t|s_t)$ which maximizes the expected return $\mathbb{E}[\sum_{t=0}^{T} r_t]$ in the POMDP by observing a series of static trajectories obtained in the same or similar environments. Each trajectory is defined as $\tau = (o_0, a_0, r_0, o_1, a_1, r_1, ..., o_T, a_T, r_T)$, and it is obtained by observing rollouts of arbitrary policies. This setup is similar to supervised learning, where models are trained from a static dataset. It is

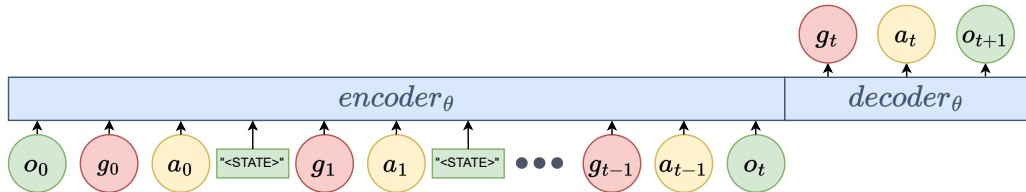

Figure 2: Our Language Decision Transformer framework. A trajectory of length $T$ is split at a random index $t \in [0, T-1]$. The model encodes the sequence of observations ($o$), goal conditions ($g$), and actions ($a$) up to time step $t$. The first $o_1$ and last $o_t$ observations are fully written, but to shorten the input sequence, the other intermediate observations are replaced by a special token. The decoder predicts the goal condition $g_t$, action to take $a_t$, and next observation $o_{t+1}$.

more difficult than online reinforcement learning since agents cannot interact with the environment to recollect more data.

**Reinforcement Learning in text-based games.** One of the main differences between traditional RL environments, such as Atari or Mujoco, and text-based environments is that both $A$ and $O$ consist of text. Therefore, due to the compositional nature of language, $A$ is significantly more complex than in common RL scenarios, where the action space is restricted to a few well-defined actions. To deal with such complexity, we model $A$, $O$ and $R$ with a large pre-trained language model: $x_i = \mathrm{LLM}(x_i | x_{1:i-1})$, where $x_i$ is the $i^{\mathrm{th}}$ text token in a text sequence of length $L$. The goal is that the LLM uses its pre-existing knowledge about the world (e.g., doors can be *opened*), to propose valid actions given an observation.

**Decision Transformers.** To perform offline learning on text-based games, we adapt the language model (particularly LongT5 (Guo et al., 2022)) to be a decision transformer (DT) (Chen et al., 2021) which abstracts reinforcement learning as a sequential modeling problem. DTs are trained with the language modeling objective on sequences of $\{g_t, o_t, a_t\}_{t=0}^T$ triples, where the goal condition $g_t$ is defined as the undiscounted sum of future rewards, or return-to-go: $g_t = \sum_{i=t}^T r_i$. Consequently, we have a model that can be conditioned on a desired goal (or return, in this case). In the following subsections, we discuss the novelties we bring to the original formulation of DTs.

## 2.2 GOAL CONDITIONING

One limitation of DTs is that the best final score of a game must be known to condition on it at the first step with $g_0$ (Chen et al., 2021). Although we have $g_0$ for the training trajectories, it is impossible to know the best target score when starting a new game. This is especially problematic for Jericho games where maximum scores vary greatly between games (Hausknecht et al., 2020).

One solution is to normalize $g_0$ during training with the maximum game score. This procedure leads to goal conditions between 0 and 1 for the training games and allows to use an initial goal condition of 1 at test time. However, this solution also assumes that we know the maximum score of every game since intermediate rewards returned by the environment $r_t$ also need to be normalized: $g_{t+1} = g_t - \frac{r_t}{\text{max score}}$. To remove the dependence on manual goal conditioning and knowledge of the best obtainable score, we take a similar approach to Lee et al. (2022) and train the model on ordered sequences of $\{o_t, g_t, a_t\}_{t=0}^T$ triples instead of $\{g_t, o_t, a_t\}_{t=0}^T$. Moving the goal condition $g_t$ after the observation $o_t$ allows us to predict the goal condition based on the current observation by modeling the joint probability of $a_t$ and $g_t$ as: $P_\theta(a_t, g_t | o_t) = P_\theta(a_t | g_t, o_t) \cdot P_\theta(g_t | o_t)$. One challenge is that sampling $g_t$ can produce low and inaccurate target returns. To mitigate this issue, we perform exponential tilting on the predicted probabilities of $g_t$. In particular we sample $g_t$ like so:

$$g_t = \mathrm{argmax}_g \big[ P_\theta(g_t | o_t) \cdot \exp(\alpha g_t) \big], \tag{1}$$

with $\alpha \geq 0$ being a hyper-parameter that controls the amount of tilting we perform. This allows us to sample high but probable target returns. We compare results with $\alpha = \{0, 1, 10, 20\}$ in Section 5.1.

Another significant advantage of predicting the goal condition $g_t$ based on $o_t$ is that we can explore various strategies of goal conditions that cannot be defined manually at inference time. We describe below the original return-to-go used by decision transformers and three novel goal condition strategies.

| | Density Estimation ($\mathcal{L}(\boldsymbol{\theta})$) | Decision Making ($\pi(\mathbf{a}|\mathbf{s};\eta)$) |
|---|---|---|
| DTs | $\log p_{\boldsymbol{\theta}}(\mathbf{a}_t \mid \mathbf{o}_t, G_t)$ | $p_{\boldsymbol{\theta}}(\mathbf{a} \mid \mathbf{s}_t, G_t)$ |
| RWR | $\exp(\eta^{-1}G_t)\log p_{\boldsymbol{\theta}}(\mathbf{a}_t \mid \mathbf{o}_t)$ | $p_{\boldsymbol{\theta}}(\mathbf{a} \mid \mathbf{s}_t)$ |
| RCP | $\log p_{\boldsymbol{\theta}}(\mathbf{a}_t \mid \mathbf{o}_t, G_t)p_{\boldsymbol{\theta}}(G_t \mid \mathbf{o}_t)$ | $p_{\boldsymbol{\theta}}(\mathbf{a} \mid \mathbf{s}_t, G)p_{\boldsymbol{\theta}}(G \mid \mathbf{s}_t)\exp(\eta^{-1}G - \kappa(\eta))$ |
| RBC | $\log p_{\boldsymbol{\theta}}(G_t \mid \mathbf{o}_t, \mathbf{a}_t)p_{\boldsymbol{\theta}}(\mathbf{a}_t \mid \mathbf{o}_t)$ | $p_{\boldsymbol{\theta}}(G \mid \mathbf{s}_t, \mathbf{a})p_{\boldsymbol{\theta}}(\mathbf{a} \mid \mathbf{s}_t)\exp(\eta^{-1}G - \kappa(\eta))$ |
| IRvS | $\log p_{\boldsymbol{\theta}}(\mathbf{a}_t, G_t \mid \mathbf{o}_t)$ | $p_{\boldsymbol{\theta}}(\mathbf{a}, G \mid \mathbf{s}_t)\exp(\eta^{-1}G - \kappa(\eta))$ |
| MB-RCP (ours) | $\log[p_{\boldsymbol{\theta}}(\mathbf{o}_{t+1} \mid \mathbf{a}_t, \mathbf{o}_t, G_t)$ $\cdot p_{\boldsymbol{\theta}}(\mathbf{a}_t \mid \mathbf{o}_t, G_t) \cdot p_{\boldsymbol{\theta}}(G_t \mid \mathbf{o}_t)]$ | $p_{\boldsymbol{\theta}}(\mathbf{a} \mid \mathbf{s}_t, G)p_{\boldsymbol{\theta}}(G \mid \mathbf{s}_t)\exp(\eta^{-1}G - \kappa(\eta))$ |

Table 1: Comparison of different policy training and action selection techniques (adapted from Piché et al. (2022)). We compare our approach with Decision Transformers (DTs) (Chen et al., 2021), Reward Weighted Regression (RWR) (Peters & Schaal, 2007; Dayan & Hinton, 1997), Reward-Conditioned Policies (RCP) (Kumar et al., 2019) (also used by Multi-Game Decision Transformers (Lee et al., 2022)), Reweighted Behavior Cloning (RBC) (Piché et al., 2019) (also used by Trajectory Transformer (TT) (Janner et al., 2021)), and Implicit RL via supervised learning (IRvS) (Piché et al., 2022). Where $\mathbf{s}$ represents the state as encoded by the model and depends on the architecture and inputs used.

**Return-To-Go (RTG):** $g_t = \sum_{i=t}^{T} r_i$, is the original strategy of the return-to-go. It is the undiscounted sum of future rewards, which will be high at the beginning of trajectories achieving a high score. These values will decrease as the agent progresses since fewer future rewards will be available in a trajectory with intermediate rewards.

**Immediate Reward (ImR):** In the setting where $g_t = r_t$, each step is conditioned on the reward observed right after the predicted action. We expect that with this goal condition method, the agent will learn what type of actions usually yield higher rewards (opening chest -vs- moving in a direction). We expect this strategy to encourage the model to get high rewards as fast as possible. However, we expect this strategy to work well only for environments with dense reward signals.

**Final Score (FinS):** $g_t = \sum_{i=0}^{T} r_i$. In this setting, each step is conditioned on the final score achieved by the agent. The final score is defined as the sum of all rewards observed during the entire trajectory. Note that, unlike all the other goal condition definitions, this score will not change over the course of a trajectory. This setting is closer to the traditional RL paradigm in which we often define rewards based on the final performance of an agent: did it win or did it lose. We expect the agent to learn to differentiate successful from unsuccessful trajectories in this setting. Since the model is not conditioned on immediate rewards, we expect it will produce longer trajectories, which can eventually achieve higher final scores.

**Average Return-To-Go (AvgRTG):** $g_t = \frac{\sum_{i=t}^{T} r_i}{(T-t)}$. In this setting, each step is conditioned on the average of all future rewards. This is also defined as the return-to-go divided by the number of steps remaining. The motivation for this goal condition is that it will capture the sparsity of rewards in a trajectory, unlike all the others. To reduce the variance in the numbers observed between different games, all goal condition numbers during training are normalized by the maximum score of the current game: $g_t = \text{int}\left[100 \cdot \frac{g_t}{\text{max score}}\right]$. At inference time, we can either manually specify goal condition numbers (assuming we know the game maximum score), or we can let the model predict those goal condition numbers with exponential tilt (more flexible). We experiment with all these goal condition definitions in our experiments and report results in Section 5.2.

### 2.3 NEXT STATE PREDICTION

To give more training signal to the model and make it more robust to stochastic environments, we also experiment with learning to predict the next observation $o_{t+1}$. Concretely, we predict $o_{t+1}$ after taking action $a_t$ in state $s_t$. Although the prediction of the next observation is not used to interact with the environment at test time, we believe that the agent will perform better if it can predict how its action will impact the world. Furthermore, predicting the next observation indirectly informs the model about the stochasticity of the environment. This technique draws parallels with the model-based paradigm in Reinforcement Learning, where the agent can predict how the environment will evolve after each action. Formally, the model estimates the following probability:

$$P_\theta(o_{t+1}, a_t, g_t|o_t) = P_\theta(o_{t+1}|a_t, g_t, o_t) \cdot P_\theta(a_t|g_t, o_t) \cdot P_\theta(g_t|o_t), \tag{2}$$

which is a type of Reward Conditioned Policy (RCP) with the additional term $P_\theta(o_{t+1}|a_t, g_t, o_t)$. We call our technique model-based reward conditioned policy (MB-RCP). We compare our formulation to prior work in Table 1. We are interested in using this additional prediction as a form of regularization and therefore treat predicting the next observation as an auxiliary loss, leading to:

$$\mathcal{L} = (1 + \lambda)^{-1}\big(L_{CE}([\hat{g}_t\hat{a}_t]; [g_t a_t]) + \lambda \cdot L_{CE}(\hat{o}_{t+1}; o_{t+1})\big), \tag{3}$$

with $L_{CE}$ being the regular cross entropy loss and $\lambda$ being a hyper-parameter set to $0.5$ in all our experiments. This weighted average prevents the model from spending too much of its representation power on the next observation prediction, as it is not strictly required to be able to interact in an environment. At inference time, only the next goal condition and next action predictions will be used. We perform an ablation study on this aspect of our approach by comparing models trained with ($\lambda = 0.5$) and without ($\lambda = 0$) this auxiliary loss and report our results in Section 5.3.

## 3 RELATED WORK

Upside-down RL (UDRL) (Schmidhuber, 2019; Kumar et al., 2019; Piché et al., 2022) poses the task of learning a policy as a supervised learning problem where an agent is conditioned on an observation and a target reward to produce an action. Instead of generating the next action for a target reward, goal conditioning methods generate trajectories conditioned on an end-goal (Ghosh et al., 2019; Paster et al., 2020). Most relevant to our work, Chen et al. (2021) recast supervised RL as a sequence modeling problem with decision transformers (DTs), but they did not examine text environments. DTs have been extended to multi-task environments by training them on multiple Atari games (Lee et al., 2022). To address the problem of modelling text-based environments Furman et al. (2022) proposed DT-BERT for question answering in TextWorld environments (Côté et al., 2018). However, the maximum number of steps in their trajectories is 50, and the environments only differ in their number of rooms and objects. Wang et al. (2022) propose ScienceWorld, a text game environment similar to TextWorld and a Text Decicion Transformer (TDT) baseline. However, their TDT model predicts only the next action based on a given expected return and the previous observation. Here we go a step further and (i) propose different conditioning methods never considered before in DTs, (ii) predict the expected return with exponential tilt rather than relying on expert knowledge to condition on it, and (iii) predict next observation after the next action prediction. In addition, we train our agents on offline trajectories across multiple games and test them on complex and realistic environments using Jericho games (Hausknecht et al., 2020) with diverse dynamics and scenarios.

Jericho is a challenging python framework composed of 33 text-based interactive fiction games (Hausknecht et al., 2020). It was initially introduced with a new Template-DQN, and compared with the Deep Reinforcement Relevance Network (DRRN) (He et al., 2016). However, both methods are trained online, which requires an expensive simulator and requires domain-specific knowledge, such as the set of possible actions. Yao et al. (2020) proposed CALM, extending DRRNs to solve the problem of it needing to know the set of possible actions in advance. They use a GPT-2 (Radford et al., 2019) language model to generate a set of possible candidate actions for each game state. Then, they use an RL agent to select the best action among the (top-k=30) generated ones.

One of the main challenges of leveraging language models to solve Jericho games is to encode the full context of the game trajectory. As such, KG-A2C (Ammanabrolu & Hausknecht, 2020) and Q*BERT (Ammanabrolu et al., 2020) use a knowledge graph to represent the environment state at each step and learn a Q-value function. SHA-KG (Xu et al., 2020) uses graph attention network (Veličković et al., 2018) to encode the game history and learn a value function. RC-DQN (Guo et al., 2020) uses a reading comprehension approach by retrieving relevant previous observations, encoding them with GRUs (Cho et al., 2014), and learning a Q-value function. DBERT-DRRN (Singh et al., 2021) leverages a DistilBERT to encode state and action and feed it to an MLP to learn a Q-value function. XTX (Tuyls et al., 2022) re-visits different frontiers in the state space and performs local exploration to overcome bottleneck states and dead-ends. CBR (Atzeni et al., 2022) stores previous interactions in memory and leverages a graph attention network (Veličković et al., 2018) to encode the similarity between states. The above previous methods are online-based RL, thus suffering from sample inefficiencies. Here, we take a simpler approach by leveraging long-context transformers like LongT5 (Guo et al., 2022) to model the sequence of state observations, target goal scores, and actions of past game trajectories as a sequence of tokens. Then, given a state observation, we leverage exponential tilt (Piché et al., 2022; Lee et al., 2022) to produce the action with the best possible target

goal score. We find that our LDT approach is effective enough to outperform all previous methods that we have examined on Jericho games.

Other prior work examining text based agents and leveraging LLMs include: The SayCan work of Ahn et al. (2022) using LLMs as a value functions in a reinforcement learning setup for completing tasks in a real world robotics setting; the ReAct work of Yao et al. (2023) examines a prompt based few shot in-context learning solution based on a PaLM-540B model; and Reflexion (Shinn et al., 2023) converts feedback from the environment into natural language sentences used in language based form of reinforcement learning.

# 4 EXPERIMENTAL SETUP

**The Jericho Engine**

Jericho[1] is a well-known Python framework that consists of 33 text-based interactive fiction games that are challenging learning environments (Hausknecht et al., 2020). Developers manually create them, each having its own way of defining the rules and goals for each game, making the games quite diverse. Text adventure games are challenging on their own because of their combinatorially large action space and sparse rewards. Usually, text adventure games have a large action vocabulary (around 2000 words on average), and each action is made of multiple words (1 to 4 on average). This makes the action space as big as $2000^4 = 1.6 \times 10^{13}$. To alleviate this issue, the Jericho benchmark provides a list of valid actions for each state. However, this makes the environment much slower as the game engine validates all possible actions against the simulator. In addition, the action space becomes dynamic as it changes from state to state. The above challenge in combination with extremely sparse rewards makes text adventure games very challenging for current RL methods. For brevity[2], we focus on 5 of the hardest Jericho games belonging to the Zork Universe: *enchanter*, *sorcerer*, *spellbrkr*, *spirit*, and *ztuu*. We report in Appendix B results on all 33 Jericho games. We generate trajectories for each of these games and train our model on the collection of all trajectories from all games.

**Data Collection**

Jericho provides one human walkthrough trajectory per game that achieves the maximum score. However, since some games are stochastic, every walkthrough is only valid for a specific default seed when initializing the game. To obtain a more diverse dataset with incorrect or partially correct trajectories, we propose to generate trajectories by following the walkthrough trajectories for some steps and then deviating from them. Concretely, to collect a large number of trajectories with different performances we follow the walkthrough trajectory for $X\%$ of its total number of steps and then take 100 additional random steps. We repeat that procedure 10 times for each $X \in [0, 5, 10, ..., 85, 90, 95]$. When $X = 0\%$, this is the same as a fully random trajectory. When $X = 95\%$, the agent follows the walkthrough path for $95\%$ of the steps and then takes 100 random steps. This results in a collection of 201 trajectories, including 1 original walkthrough for each game. Note that we also tried to include TDQN and DRRN trajectories trained on individual games, but these agents did not bring any significant information gain in our collection of trajectories. To not overfit on the default seed for each game, we ran the same procedure on 5 different seeds. This resulted in 1,005 trajectories of various lengths and qualities for each game. Note that only 1 of those obtain a 100% final score by following the walkthrough actions given by Jericho. We report in Appendix C the normalized scores (Figure 5) and lengths (Figure 6) observed in the collection of trajectories collected for each game. The top part of Figure 1 illustrates the data generation procedure.

**Sequence Definition**

To train an encoder-decoder architecture, trajectories are split between input and output sequences after a random number of steps $t \in [0, T-1]$ . The input sequence is then defined as $[o_0, g_0, a_0, o_1, ..., g_{t-1}, a_{t-1}, o_t]$ and the output sequence as $[g_t, a_t, o_{t+1}]$ (also depicted in Figure 2). Each of these $\{o_t, g_t, a_t\}_{t=0}^T$ elements are represented in natural language text (described below) and concatenated together to form input/output text sequence pairs.

---

[1] https://github.com/microsoft/jericho

[2] These games are also used in previous works, allowing us to compare our results. Most prior work has only evaluated on a subset of the 33 environments. We report in Appendix B results on all 33 games.

$a_t$: intermediate actions are written as returned by agents playing the game, with the addition of special token delimiters: "`Action: {a_t} `".

$g_t$: goal conditions are computed with one strategy among the ones described in Section 2.2 based on the list of intermediate rewards returned by the environment. Each goal condition is written in text like this: "`GC: {g_t} `".

$o_t$: state observations are defined by multiple state characteristics available to Jericho games: (i) candidate actions available, (ii) the message returned by the game engine, (iii) the description of the current room, and (iv) the current inventory of the agent. Each observation is written in text like this: "`Actions: {cand}  State: {msg}  Description: {desc}  Inventory: {inv} `", with `{cand}`, `{msg}`, `{desc}` and `{inv}` being the list of candidate actions, the game message, the description of the current room, and the inventory of the player respectively.

However, as some game trajectories contain hundreds of steps, the current definition of $\{o_t, g_t, a_t\}_{t=0}^T$ triples can make input sequences as long as tens of thousands of tokens. To shorten input sequences, we replaced state observations $o_t$ to be a single placeholder token "`<STATE>`" for all intermediate observations except the first ($o_0$) and current one ($o_t$) as depicted in Figure 2.

## 5 EXPERIMENTAL RESULTS

Since Jericho games have long storylines, we leverage LongT5 (Guo et al., 2022), a text-to-text Transformer with a wide attention span. We use the pre-trained `LongT5-base` model as hosted by HuggingFace[3] (Wolf et al., 2020) in all experiments as the base for our encoder-decoder architecture. We then fine-tuned the model for multiple epochs on the generated trajectories from Section 4. The hyperparameter settings can be found in Appendix A.

For each game, we initialize its environment with a random seed. We let the model predict the next goal condition and action at each step. The agent performs the predicted action, leading to the next observation in the environment. The model uses this observation as context for the next step. We run these steps in a cycle until we reach the end of the game and compute the final score. The game ends when the agent reaches the final state, or the model generates an invalid sequence. We repeat this process on 5 random seeds and take the average final score. The bottom part of Figure 1 illustrates the training and evaluation process.

### 5.1 THE EFFECT OF EXPONENTIAL TILT

In this section, we fine-tuned our model with the loss function described in Equation 3 on all generated trajectories split into input and output pairs (Section 4). The model was trained with the regular return-to-go goal condition ($g_t = \sum_{i=t}^T r_i$) and with $\lambda = 0.5$ for the auxiliary loss of predicting $o_{t+1}$. We tested the model on all games[4], normalized the obtained score based on the maximum human score for each game, and recorded the average across games and 5 random seeds for each game.

To measure the effect of exponential tilt, the predicted $g_t$ were sampled according to Equation 1 with $\alpha = 0, 1, 10, 20$ ("Predicted GC / alpha=$\alpha$" in Figure 3). We also evaluated the model with the goal condition being manually given ("Optimal GC" in Figure 3) at each step. In the first step, the model is conditioned with $g_0 = 100\%$, and at every next step $g_t$ is reduced by the amount of observed reward: $g_{t+1} = g_t - \frac{r_t}{\text{max score}}$. This "Optimal GC" evaluation assumes we know the game's maximum score. We aim to achieve similar performance by simply predicting $g_t$ instead of manually defining it.

We report in Figure 3 the normalized score averaged across games for each method of predicting $g_t$ at different training stages of the model. During training the model is exposed to trajectories of various performances (detailed in Figure 5), so without any exponential tilt the model will output the most probable goal condition based on what it observed during training, which is less than ideal (solid red "Predicted GC / alpha=0" line). However, as we prioritize high numerical return-to-go over their likelihood ($\alpha$ increasing), the model's performance is getting closer to the "Optimal GC"

---

[3]https://huggingface.co/google/long-t5-tglobal-base

[4]games seen at training time: *enchanter*, *sorcerer*, *spellbrkr*, *spirit*, *ztuu*. We report in Appendix B results on all 33 games.

performance, with $\alpha = 20$ (solid orange line) being on par with the model that was manually given the "optimal" goal condition. In realistic scenarios, we do not have the optimal goal condition when starting a new game. In addition, predicting the goal condition offers greater flexibility in the design of goal conditions. We can now explore conditioning methods that would be impossible to define manually during run time. This is exactly what we explore in the next section. Overall, these results demonstrate that: (1) the numerical value of the goal condition has indeed an effect on the quality of the next generated answer, and (2) it is possible to recover the same performance as the "optimal" goal conditioning by increasing the amount of exponential tilt without knowing the game's max score.

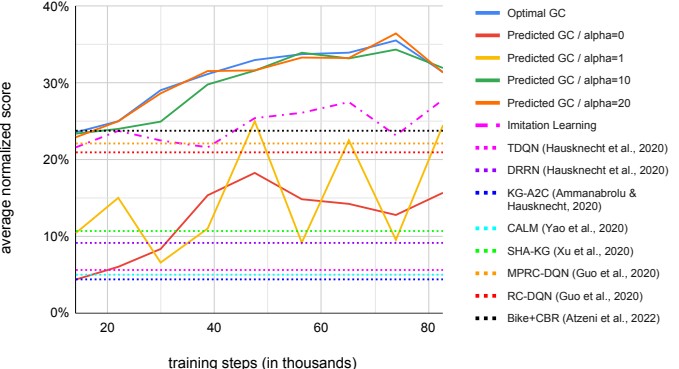

Furthermore, we report in Figure 3 the average performance of previous works on the same set of games (dotted horizontal lines). Our offline method with exponential tilt beats previous methods with very little training when $\alpha = 10$ or 20. However, since all previous methods were trained on each game in an online RL fashion, to fairly compare our approach we also trained an imitation learning (IL) baseline on human trajectories only (semidotted pink line). As expected, this offline RL baseline performs better than

Figure 3: Average normalized score across different Jericho games (enchanter, sorcerer, spellbrkr, spirit, ztuu) with various amounts of exponential tilt ("*Predicted GC*" lines). We also report the performance of a model being conditioned on the optimal goal according to each game's maximum score ("*Optimal GC*" line). The average normalized score of various baselines is depicted in dotted lines.

our approach without exponential tilt as it was trained on better trajectories, but as we increase exponential tilt ($\alpha = 10$ or 20), our method outperforms the IL strategy. This can be explained by the diversity of interactions our agent saw during training compared to IL. This difference in performance also illustrates the stochasticity of the environments, as the IL baseline would perform close to 100% on deterministic games.

## 5.2 THE EFFECT OF GOAL CONDITIONING STRATEGIES

We fine-tuned 4 models with the loss function in Equation (3) on all generated trajectories split into input and output pairs (Section 4). Each model was trained with a different goal condition (Section 2.2) and with $\lambda = 0.5$ for the auxiliary loss predicting $o_{t+1}$. We tested the models on all games[5] after 31.4k training steps and recorded the average score across 5 random seeds per game. In these experiments, the model generates goal conditions because at inference time, unlike with return-to-go (RTG), we cannot compute the immediate next reward (ImR) and the average return-to-go (AvgRTG), even if we know the game maximum score. To provide the immediate next reward condition, we need to know at each step the maximum achievable reward among all candidate actions, which is infeasible in practice. To provide the average RTG condition, we need to know the number of steps remaining after each state, which is infeasible in practice. Fortunately, our model can generate goal conditions while leveraging the exponential tilt for producing better trajectories. All models in these experiments were evaluated by sampling $g_t$ according to Equation 1 with $\alpha = 10$.

Table 2 reports the average score, standard deviation, and best score obtained on each game across 5 random seeds for all goal conditioning methods. Overall, these results show that the classical return-to-go conditioning method yields weaker performance than other methods in all environments. However, the best strategy depends on the game which can vary between ImR, FinS, or AvgRTG. These results further motivate the advantages of generating goal conditions that cannot be computed at runtime such as ImR and AvgRTG.

---

[5]Games seen at training time: *enchanter*, *sorcerer*, *spellbrkr*, *spirit*, *ztuu*. We report in Appendix B results on all 33 games.

| GC = | Return-To-Go | | | Immediate Reward | | | Final Score | | | Avg.RTG | | |
|---|---|---|---|---|---|---|---|---|---|---|---|---|
| | avg. | stdev. | best | avg. | stdev. | best | avg. | stdev. | best | avg. | stdev. | best |
| enchanter | 45.0 | 0.0 | 45.0 | **235.0** | 0.0 | 235.0 | 231.0 | 56.7 | 280.0 | 175.0 | 0.0 | 175.0 |
| sorcerer | 124.0 | 90.6 | 235.0 | 112.0 | 75.9 | 205.0 | 124.0 | 90.6 | 235.0 | **132.0** | 100.4 | 255.0 |
| spellbrkr | 31.0 | 7.4 | 40.0 | 31.0 | 7.4 | 40.0 | 25.0 | 0.0 | 25.0 | **40.0** | 0.0 | 40.0 |
| spirit | 18.4 | 3.9 | 26.0 | 22.4 | 8.7 | 38.0 | **26.0** | 15.4 | 56.0 | 5.6 | 0.8 | 6.0 |
| ztuu | 73.0 | 7.5 | 85.0 | **75.0** | 9.5 | 90.0 | **75.0** | 9.5 | 90.0 | **75.0** | 9.5 | 90.0 |
| Norm. Avg. | 26.7% | | 35.9% | 36.4% | | 45.9% | **36.6%** | | 50.0% | 33.7% | | 42.8% |

Table 2: Average and best score obtained on each game across 5 random seeds for each goal condition (GC) variation. The bottom row is the normalized average based on the best human score.

## 5.3 THE EFFECT OF PREDICTING THE NEXT OBSERVATION

Here we analyze the effect of predicting the next observation $o_{t+1}$ as part of the loss function. We fine-tuned another 4 models, each with a different goal condition similar to the above section, but with the loss function described in Equation 3 with $\lambda = 0.0$ for the auxiliary loss of predicting $o_{t+1}$. We tested the models on the same set of games after 31.4k training steps and recorded the average score across 5 random seeds for each game. To compare the effect of the auxiliary loss, we averaged the scores across all goal conditioning methods.

| | $\lambda = 0.0$ | | | $\lambda = 0.5$ | | |
|---|---|---|---|---|---|---|
| | avg. | stdev. | best | avg. | stdev. | best |
| enchanter | 138.8 | 55.8 | 235.0 | **171.5** | 81.9 | 280.0 |
| sorcerer | 79.5 | 39.4 | 130.0 | **123.0** | 90.1 | 255.0 |
| spellbrkr | 25.8 | 3.3 | 40.0 | **31.8** | 7.5 | 40.0 |
| spirit | 10.2 | 9.8 | 36.0 | **18.1** | 11.9 | 56.0 |
| ztuu | 55.8 | 33.1 | 90.0 | **74.5** | 9.1 | 90.0 |
| Norm. Avg. | 24.7% | | 42.0% | **33.3%** | | 52.1% |

Table 3: Average and best score obtained on each game across 5 random seeds and 4 goal conditioning methods, with ($\lambda = 0.5$) and without ($\lambda = 0.0$) the auxiliary loss on the prediction of the next observation $o_{t+1}$. The bottom row is the normalized average based on the best human score.

Table 3 reports the average score, standard deviation, and best score obtained on each game over 20 runs (5 random seeds × 4 goal conditioning methods) for models trained with ($\lambda = 0.5$) and without ($\lambda = 0.0$) the auxiliary loss on the predicted next observation $o_{t+1}$. In all games, models trained to predict the next observation $o_{t+1}$ resulting from the predicted action $a_t$ and goal condition $g_t$ perform better than models trained to only predict the goal condition $g_t$ and next action $a_t$. Overall, these results show that our proposed model-based reward-conditioned policy (MB-RCP) learning objective yields stronger performance than the classical reward-conditioned policy (RCP) objective.

## 6 CONCLUSION

In this work, we have proposed Language Decision Transformers (LDTs) as an offline reinforcement learning method for interactive text environments, and we have performed experiments using the challenging text-based games of Jericho. LDTs are built from pre-trained LLMs followed by training on multiple games simultaneously to predict: the trajectory goal condition, the next action, and the next observation. We have shown that by using exponential tilt, LDT-based agents get much better performance than otherwise. In fact, the model obtains similar performance as if it was conditioned on the optimal goal, despite the fact that in most realistic scenarios, we do not have access to that optimal goal condition. We have also explored different conditioning methods and observed that the traditional return-to-go was the weakest strategy. Finally we have seen that training the model to predict the next observation as an auxiliary loss improves performance. For future work, we plan on extending this framework to multiple and more diverse games and environments. We hope this work can provide a missing piece to the substantial advances in the application of large language models in the context of real-world interactive task-oriented dialogues.

**Limitations.** One limitation of this work is that we did not spend an extensive amount of effort in building high-quality online RL agents to train our offline agent. This is intended because we use Jericho games as a proxy for real-world chat agents helping people solve problems, and in such environments training a descent online agent is impractical as people would find it very challenging to interact with live RL agents. Another limiting factor of this work is the fact that due to computing resource limitations, intermediate states were replaced with fixed tokens. Earlier experiments tried to encode them with a frozen encoder network but results were inconclusive. Further research in long-context Transformers will eventually alleviate this limitation. Eventually, at current capacity, our models are unable to generalize to unseen games in zero-shot settings.

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

## A   TRAINING DETAILS

| | |
|---:|:---|
| optimizer | Adafactor |
| learning rate | 1e-4 |
| precision | 32 |
| strategy | deepspeed-stage-2 |
| batch size | 2 |
| gradient accumulation | 8 |
| effective batch size | 16 |
| max input length | 4096 |
| max output length | 1024 |
| base model | LongT5-base with Transient Global Attention (google/long-t5-tglobal-base) |
| training data | 201 trajectories per game per seed × 5 games × 5 seed = 5,025 sequences |
| gpu requirements | 2 × NVIDIA A100-SXM4-80GB |
| **library requirements** | |
| python | 3.8.13 |
| deepspeed | 0.6.3 |
| numpy | 1.23.4 |
| torch | 1.13.0 |
| pytorch-lightning | 1.5.10 |
| transformers | 4.20.0 |
| jericho | 3.1.0 |

## B   RESULTS ON ALL GAMES

This section reports the results of models trained on all game trajectories at the same time (33 games ×1005 trajectories). Models are then evaluated on each individual games.

### B.1   THE EFFECT OF EXPONENTIAL TILT

Similarly as in Section 5.1, we fine-tuned our model with the loss function described in Equation 3 on all generated trajectories split into input and output pairs (Section 4). The model was trained with the regular return-to-go goal condition ($g_t = \sum_{i=t}^{T} r_i$) and with $\lambda = 0.5$ for the auxiliary loss of predicting $o_{t+1}$. We tested the model on all games, normalized the obtained score based on the maximum human score for each game, and recorded the average across games and 5 random seeds for each game.

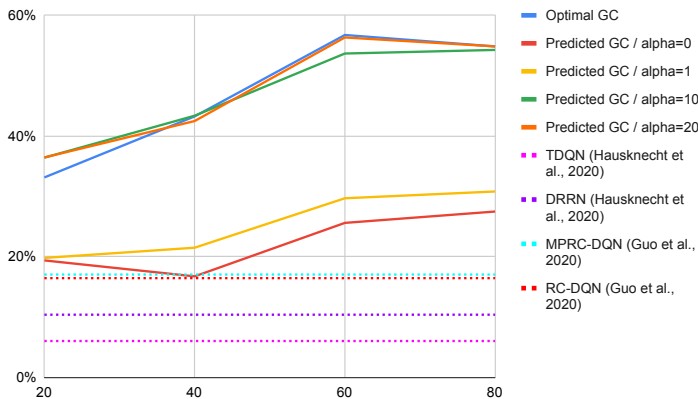

Figure 4: Average normalized score across all Jericho games with various amounts of exponential tilt ("*Predicted GC*" lines). We also report the performance of a model being conditioned on the optimal goal according to each game's maximum score ("*Optimal GC*" line). The average normalized score of various baselines is depicted in dotted lines.

We can observe the same conclusions as in Section 5.1: (1) the numerical value of the goal condition has indeed an effect on the quality of the next generated answer, (2) it is possible to recover the same performance as the "optimal" goal conditioning by increasing the amount of exponential tilt without knowing the game's maximum score, and (3) our offline method beats previous methods with very little training. Note: fewer baselines are reported than in Section 5.1 because many previous work do not report their performance on so many different environments.

## B.2  THE EFFECT OF GOAL CONDITIONING STRATEGIES

Similarly as in Section 5.2, we fine-tuned 4 models with the loss function described in Equation 3 on all generated trajectories split into input and output pairs (Section 4). Each model was trained with a different goal condition (Section 2.2) and with $\lambda = 0.5$ for the auxiliary loss of predicting $o_{t+1}$. We tested the models on all games after 60k training steps and recorded the average score across 5 random seeds for each game.

| GC = | Return-To-Go | | | Imediate Reward | | | Final Score | | | Avg. Return-To-Go | | |
|---|---|---|---|---|---|---|---|---|---|---|---|---|
| | avg. | stdev. | best | avg. | stdev. | best | avg. | stdev. | best | avg. | stdev. | best |
| **905** | **0** | 0.00 | 0.00 | **0** | 0.00 | 0.00 | **0** | 0.00 | 0.00 | **0** | 0.00 | 0.00 |
| acorncourt | **30** | 0.00 | 30.0 | **30** | 0.00 | 30.00 | **30** | 0.00 | 30.00 | 20 | 10.00 | 30.00 |
| advent | **155.5** | 41.81 | 223.0 | 144.3 | 27.82 | 180.00 | 152.9 | 42.76 | 235.00 | 118.5 | 33.12 | 150.00 |
| adventureland | 21.0 | 5.42 | 35.0 | 28 | 11.71 | 42.00 | **59.2** | 22.82 | 74.00 | 38.7 | 24.43 | 67.00 |
| afflicted | 48.8 | 26.57 | 75.0 | 37.5 | 37.50 | 75.00 | **58.5** | 20.01 | 75.00 | 41 | 19.47 | 55.00 |
| anchor | 14.1 | 7.27 | 23.0 | **23.6** | 2.80 | 25.00 | 14.9 | 10.39 | 25.00 | 17.2 | 7.85 | 25.00 |
| awaken | 34.5 | 4.72 | 45.0 | 21 | 9.43 | 30.00 | **35** | 0.00 | 35.00 | 0 | 0.00 | 0.00 |
| balances | **30** | 0.00 | 30.0 | **30** | 0.00 | 30.00 | 15 | 15.00 | 30.00 | 20 | 10.00 | 30.00 |
| deephome | 130.3 | 45.59 | 191.0 | 87.4 | 86.61 | 181.00 | **149.1** | 26.25 | 181.00 | 48.3 | 48.38 | 157.00 |
| detective | 190.0 | 170.00 | 360.0 | **360** | 0.00 | 360.00 | **360** | 0.00 | 360.00 | **360** | 0.00 | 360.00 |
| dragon | 3.4 | 39.89 | 63.0 | 7.9 | 7.50 | 17.00 | 10 | 9.11 | 20.00 | **11.3** | 6.65 | 20.00 |
| enchanter | 191.0 | 51.47 | 235.0 | 145 | 88.91 | 280.00 | **214** | 42.00 | 280.00 | 117.5 | 117.50 | 235.00 |
| gold | 32.7 | 14.72 | 48.0 | 33.3 | 8.53 | 48.00 | **36.9** | 7.83 | 48.00 | 34.2 | 8.29 | 51.00 |
| inhumane | 0.0 | 0.00 | 0.0 | 70 | 0.00 | 70.00 | 75 | 15.00 | 90.00 | 45 | 45.00 | 90.00 |
| jewel | **72.5** | 2.50 | 75.0 | 44.5 | 25.50 | 70.00 | 43 | 27.00 | 70.00 | 52 | 18.00 | 70.00 |
| karn | 32.0 | 26.94 | 65.0 | 37 | 25.32 | 75.00 | 37 | 25.32 | 75.00 | 17 | 19.00 | 40.00 |
| library | 14.0 | 13.00 | 27.0 | 25 | 5.00 | 30.00 | 29 | 1.00 | 30.00 | 24 | 4.00 | 28.00 |
| ludicorp | **116.5** | 10.50 | 127.0 | 29.5 | 19.50 | 49.00 | 80.5 | 7.50 | 88.00 | 54 | 34.00 | 88.00 |
| moonlit | **0.5** | 0.50 | 1.0 | **0.5** | 0.50 | 1.00 | 0 | 0.00 | 0.00 | 0 | 0.00 | 0.00 |
| omniquest | 20.0 | 5.00 | 25.0 | 25 | 0.00 | 25.00 | 25 | 0.00 | 25.00 | 15 | 10.00 | 25.00 |
| pentari | **39.0** | 9.95 | 45.0 | 20 | 0.00 | 20.00 | 29.5 | 10.83 | 45.00 | 27.5 | 11.46 | 45.00 |
| reverb | 35.0 | 5.00 | 40.0 | 25 | 18.00 | 43.00 | 25 | 15.00 | 40.00 | **39.5** | 1.50 | 40.00 |
| snacktime | 30.0 | 20.00 | 50.0 | **50** | 0.00 | 50.00 | 40 | 10.00 | 50.00 | 35 | 15.00 | 50.00 |
| sorcerer | 86.0 | 44.54 | 150.0 | 83 | 41.90 | 150.00 | **90** | 50.60 | 170.00 | 81 | 62.60 | 205.00 |
| spellbrkr | 25.0 | 0.00 | 25.0 | 26.5 | 4.50 | 40.00 | 25 | 0.00 | 25.00 | **34** | 7.35 | 40.00 |
| spirit | **14.4** | 3.47 | 19.0 | 6.4 | 4.18 | 12.00 | 8.6 | 5.85 | 17.00 | 3.8 | 2.89 | 12.00 |
| temple | 26.2 | 5.29 | 30.0 | **27.5** | 2.50 | 30.00 | 18.3 | 6.72 | 25.00 | 25 | 0.00 | 25.00 |
| tryst205 | 20.0 | 15.00 | 35.0 | 40 | 10.00 | 50.00 | **77.5** | 27.50 | 105.00 | 20 | 10.00 | 30.00 |
| yomomma | 12.0 | 12.00 | 24.0 | **12.6** | 12.60 | 26.00 | 7.5 | 7.50 | 15.00 | 7.5 | 7.50 | 15.00 |
| zenon | **12** | 8.00 | 20.0 | **12** | 8.00 | 20.00 | 9.2 | 3.60 | 12.00 | 0 | 0.00 | 0.00 |
| zork1 | **68.3** | 41.33 | 142.0 | 63.5 | 32.56 | 123.00 | 50.8 | 18.76 | 78.00 | 56.8 | 41.12 | 136.00 |
| zork3 | 2.6 | 1.20 | 5.0 | 2.6 | 1.02 | 5.00 | **3** | 1.00 | 5.00 | 1.2 | 1.33 | 4.00 |
| ztuu | 65.0 | 0.00 | 65.0 | 69.5 | 7.89 | 90.00 | 60 | 14.04 | 85.00 | **72** | 7.81 | 85.00 |
| **Avg. Norm.** | 42.35% | | 66.16% | 43.99% | | 61.00% | **45.40%** | | 59.84% | 35.37% | | 52.56% |

Table 4: Average and best score obtained on each game across 5 random seeds for each goal condition (GC) variation. The bottom row is the normalized average based on the best human score.

We can observe two things: (1) averaged over a wider set of games, all goal conditioning strategies yield similar performances, except Avg.RTG which is lower, and (2) results on *enchanter*, *sorcerer*, *spellbrkr*, *spirit* and *ztuu* are weaker than in Table 2. This is because although models are trained for twice as many steps, the training data is 6 times larger, hence the model observed 3 times less interaction from each game compared to our setting in Section 5.2.

## B.3  THE EFFECT OF PREDICTING THE NEXT OBSERVATION

Similarly as in Section 5.3, we fine-tuned another 4 models, each with a different goal condition similar to the above section, but with the loss function described in Equation 3 with $\lambda = 0.0$ for the auxiliary loss of predicting $o_{t+1}$. We tested the models on all games after 60k training steps and recorded the average score across 5 random seeds for each game. To compare the effect of the auxiliary loss, we averaged the scores across all goal conditioning methods.

| game | max | $\lambda = 0.0$ | | | $\lambda = 0.5$ | | |
|---|---|---|---|---|---|---|---|
| | | avg. | stdev. | best. | avg. | stdev. | best. |
| 905 | 1 | **0** | 0.00 | 0.00 | **0** | 0.00 | 0.00 |
| acorncourt | 30 | 25 | 8.66 | 30.00 | **30** | 0.00 | 30.00 |
| advent | 350 | 129.3 | 32.90 | 180.00 | **156.3** | 41.29 | 235.00 |
| adventureland | 100 | 29.5 | 20.00 | 67.00 | **43.95** | 23.53 | 74.00 |
| afflicted | 75 | 41.65 | 24.24 | 75.00 | **51.25** | 30.70 | 75.00 |
| anchor | 99 | 11.8 | 8.01 | 25.00 | **23.1** | 3.86 | 25.00 |
| awaken | 50 | 20.75 | 15.51 | 40.00 | **24.5** | 14.57 | 45.00 |
| balances | 50 | 17.5 | 12.99 | 30.00 | **30** | 0.00 | 30.00 |
| deephome | 300 | **119.3** | 39.94 | 181.00 | 88.25 | 85.32 | 191.00 |
| detective | 360 | 275 | 147.22 | 360.00 | **360** | 0.00 | 360.00 |
| dragon | 25 | 2.6 | 2.85 | 6.00 | **13.7** | 28.91 | 63.00 |
| enchanter | 400 | 104.25 | 69.31 | 175.00 | **229.5** | 57.23 | 280.00 |
| gold | 100 | 25.95 | 7.49 | 42.00 | **42.6** | 4.51 | 51.00 |
| inhumane | 90 | 32.5 | 32.69 | 70.00 | **62.5** | 37.00 | 90.00 |
| jewel | 90 | 34.75 | 21.46 | 70.00 | **71.25** | 2.17 | 75.00 |
| karn | 170 | 11.5 | 7.92 | 20.00 | **50** | 22.69 | 75.00 |
| library | 30 | 19.75 | 11.45 | 30.00 | **26.25** | 3.77 | 30.00 |
| ludicorp | 150 | 65.75 | 33.50 | 106.00 | **74.5** | 42.13 | 127.00 |
| moonlit | 1 | 0 | 0.00 | 0.00 | **0.5** | 0.50 | 1.00 |
| omniquest | 50 | 17.5 | 8.29 | 25.00 | **25** | 0.00 | 25.00 |
| pentari | 70 | 28.5 | 12.16 | 45.00 | **29.5** | 10.83 | 45.00 |
| reverb | 50 | 24.25 | 15.79 | 40.00 | **38** | 4.95 | 43.00 |
| snacktime | 50 | 27.5 | 14.79 | 50.00 | **50** | 0.00 | 50.00 |
| sorcerer | 400 | 76 | 40.52 | 150.00 | **94** | 57.70 | 205.00 |
| spellbrkr | 280 | 26.5 | 4.50 | 40.00 | **28.75** | 6.50 | 40.00 |
| spirit | 250 | 5.45 | 5.57 | 19.00 | **11.15** | 4.40 | 17.00 |
| temple | 35 | 21 | 6.12 | 25.00 | **27.5** | 2.50 | 30.00 |
| tryst205 | 320 | 33.75 | 18.50 | 50.00 | **45** | 35.88 | 105.00 |
| yomomma | 34 | 0 | 0.00 | 0.00 | **19.8** | 4.82 | 26.00 |
| zenon | 20 | 6.6 | 6.10 | 16.00 | **10** | 8.72 | 20.00 |
| zork1 | 350 | 52.4 | 25.48 | 98.00 | **67.3** | 41.63 | 142.00 |
| zork3 | 7 | 1.95 | 1.36 | 5.00 | **2.75** | 1.18 | 5.00 |
| ztuu | 100 | 62 | 9.85 | 85.00 | **71.25** | 7.89 | 90.00 |
| **Average Normalized** | | 32.78% | | 54.45% | **50.78%** | | 73.78% |

Table 5: Average and best score obtained on each game across 5 random seeds and 4 goal conditioning methods, with ($\lambda = 0.5$) and without ($\lambda = 0.0$) the auxiliary loss on the prediction of the next observation $o_{t+1}$. The bottom row is the normalized average based on the best human score.

We can observe the same conclusions as in Section 5.3: models trained to predict the next observation $o_{t+1}$ resulting from the predicted action $a_t$ and goal condition $g_t$ perform better than models trained to only predict the goal condition $g_t$ and next action $a_t$. Overall, these results show that our proposed model-based reward-conditioned policy (MB-RCP) learning objective yields stronger performance than the classical reward-conditioned policy (RCP) objective.

# C  TRAJECTORIES STATISTICS

In this section, we report the normalized scores (Figure 5) and lengths (Figure 6) observed in the collection of trajectories collected for each game as described in Section 4.

## C.1  TRAJECTORIES SCORES

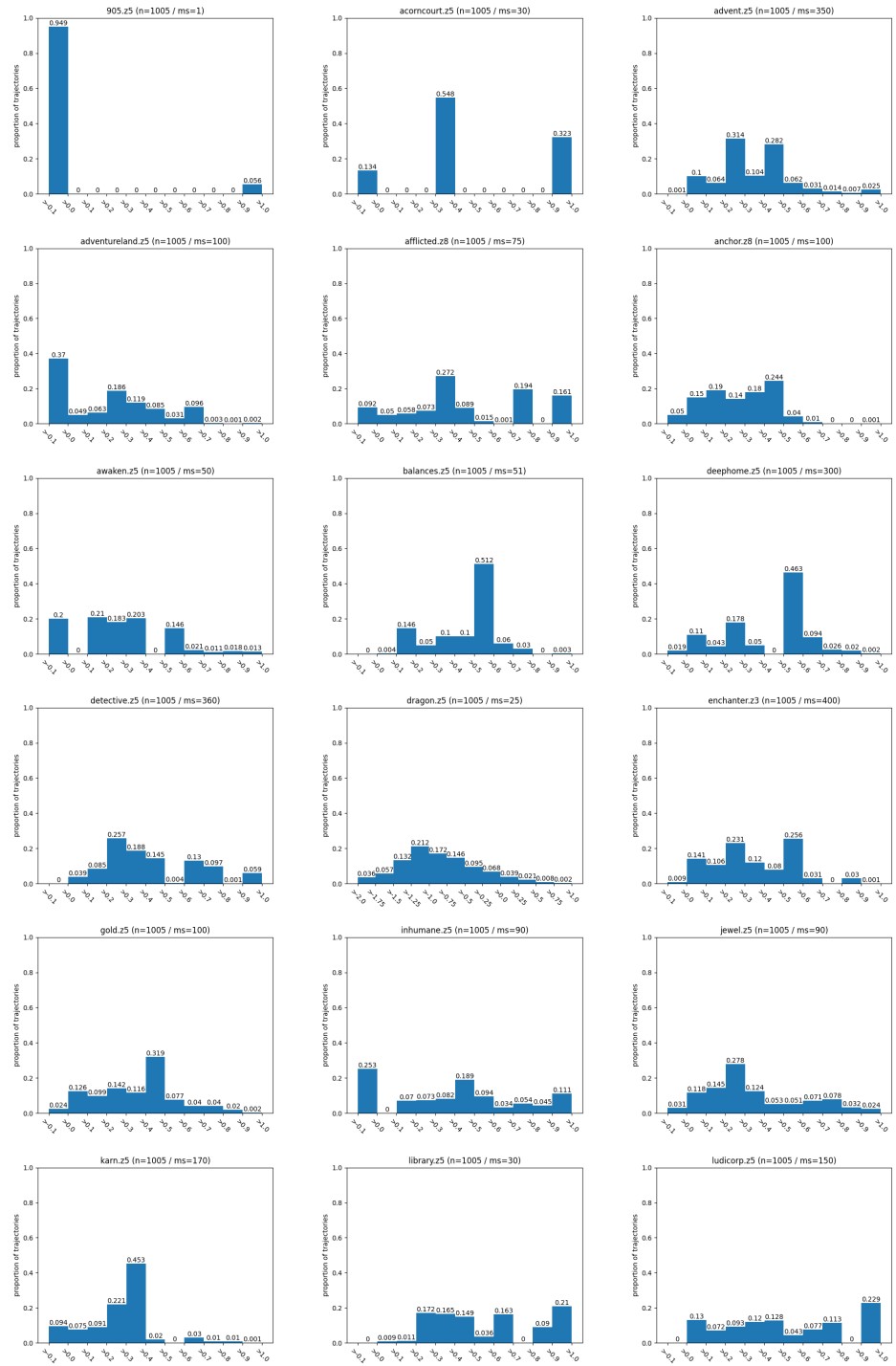

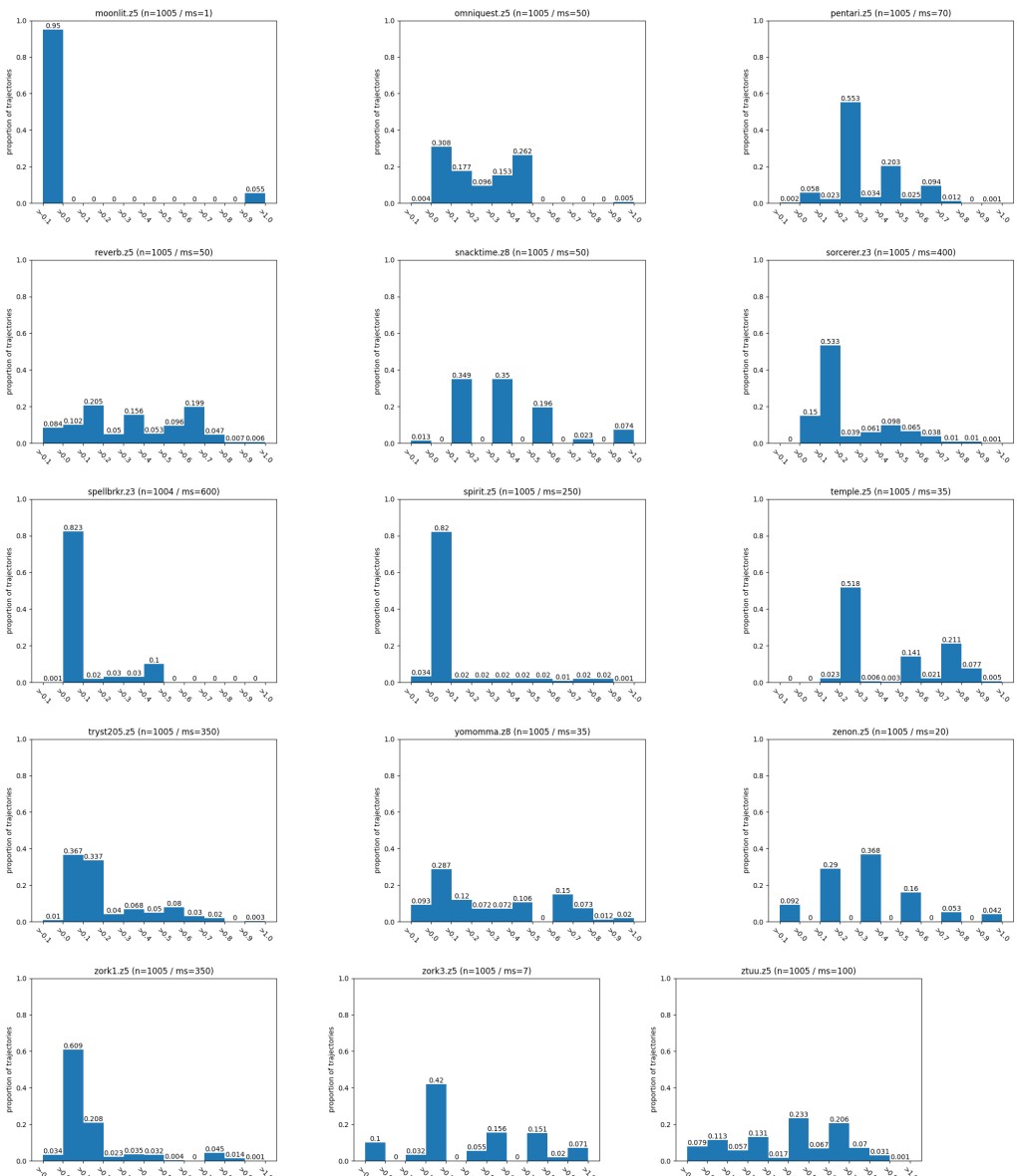

Figure 5: Proportion of trajectory normalized scores. In each sub-figure title, $n$ is the number of trajectories and $ms$ is the maximum score. The X-axis is the normalized score the trajectory achieves. The Y-axis is the proportion of trajectories finishing with that score.

## C.2    TRAJECTORIES LENGTHS

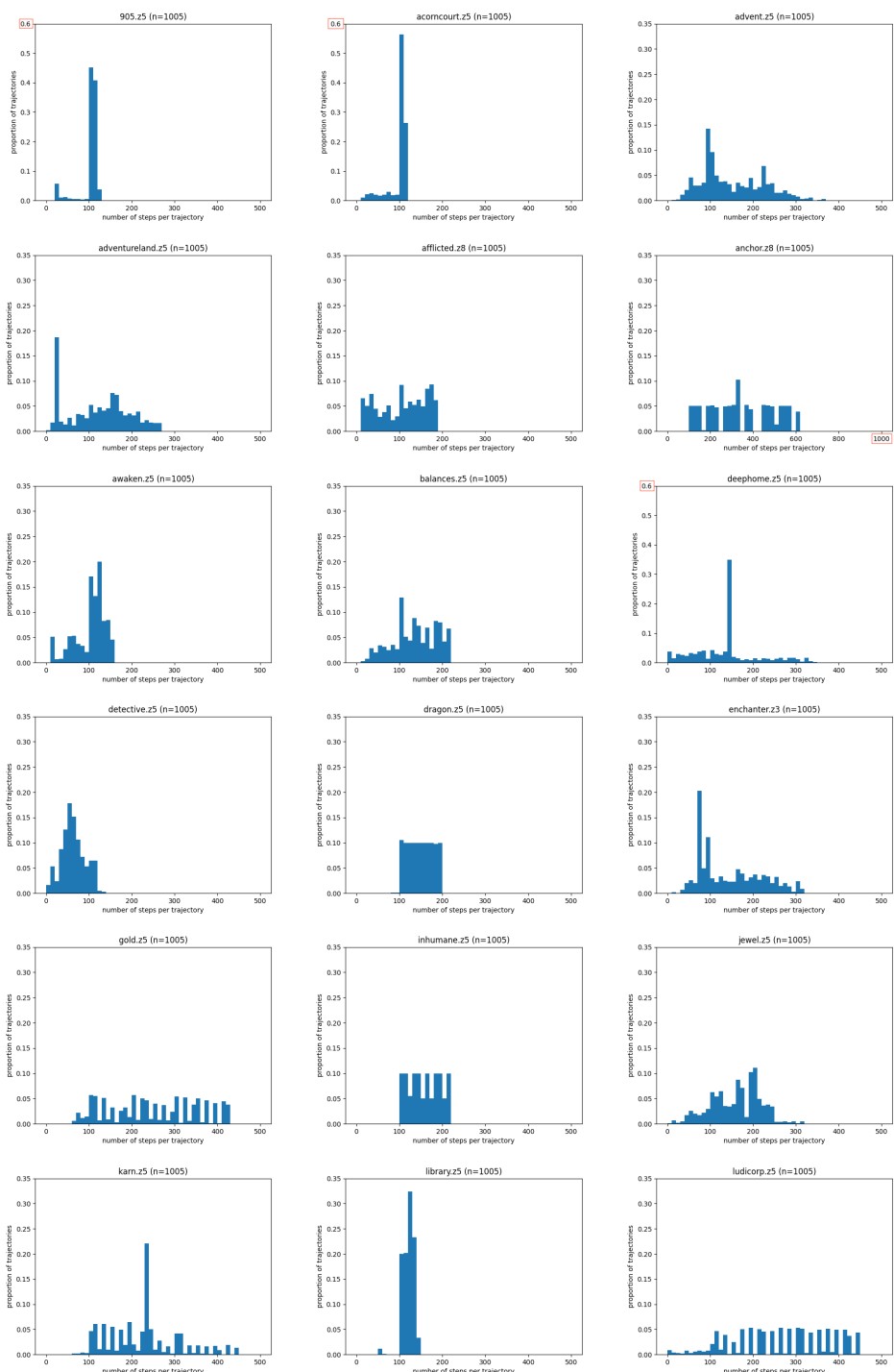

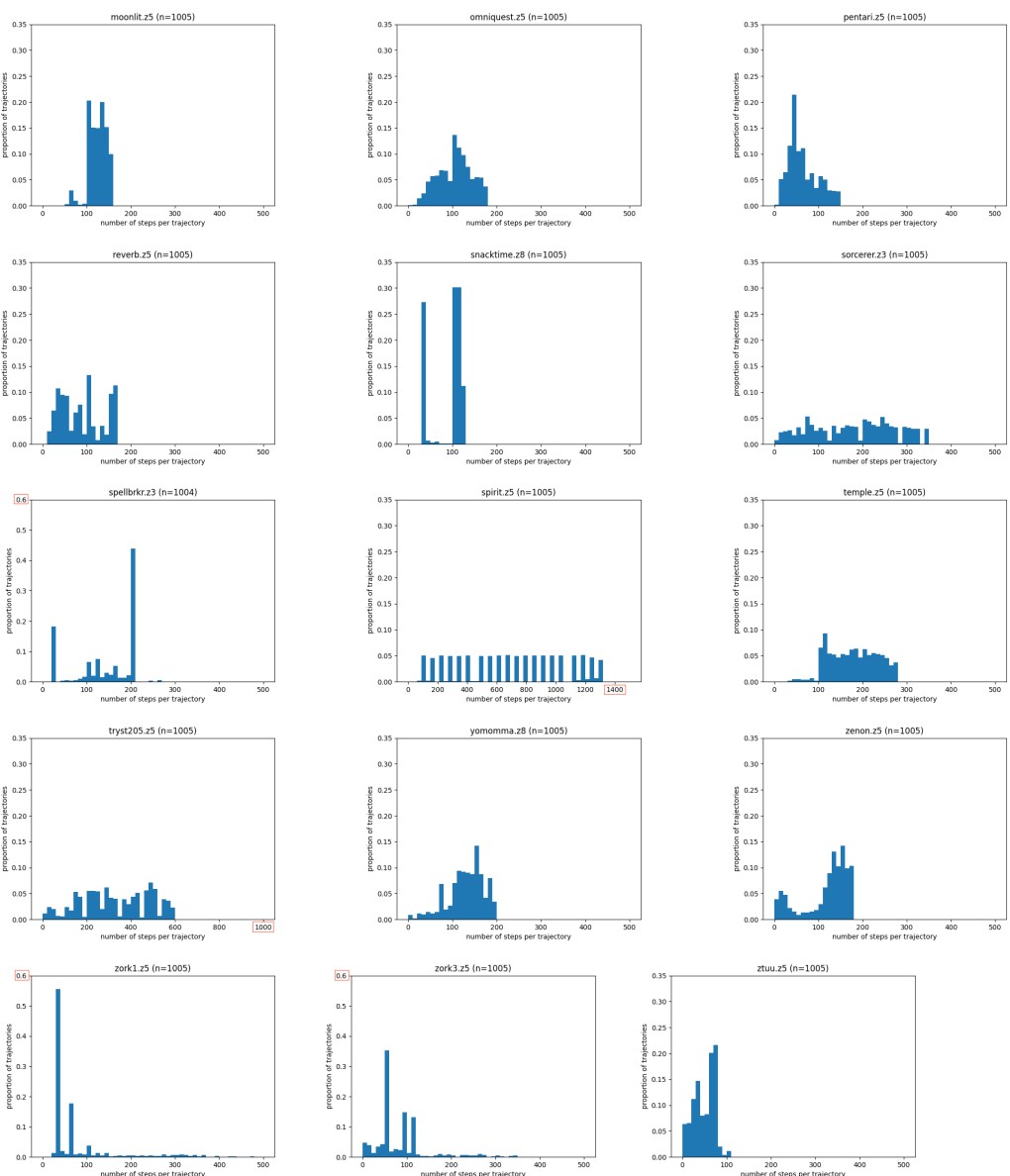

Figure 6: Proportion of trajectory lengths. In each sub-figure title, $n$ is the number of trajectories. The X-axis is the number of steps in a trajectory. The Y-axis is the proportion of trajectories of that length.

