# OpenReview forum: "Language Decision Transformers with Exponential Tilt for Interactive Text Environments"
_ICLR.cc/2024/Conference — ICLR 2024 Conference Withdrawn Submission_

### Official Review · Reviewer_NX9B · 2023-10-31

**Soundness:** 3 good
**Presentation:** 3 good
**Contribution:** 1 poor
**Rating:** 3
**Confidence:** 3

**Summary:**

The authors develop a decision transformer for text-based games. They introduce further enhancements in the form of alternative goal conditioning and next state prediction. On text games based on the Jericho engine, the proposed technique perform 10% better than previous baselines on the hardest environments, and up to 30% better on average.

**Strengths:**

The empirical results support the objectives by demonstrating improved performance in text-based games using the proposed approach.

**Weaknesses:**

The impact of this work is limited:
1. the technical contribution of this work are mostly an amalgamation of prior techniques, for instance next state prediction (Dreamer 2019: https://arxiv.org/pdf/1912.01603.pdf), goal conditioning (Lee 2022 as cited by authors).
2. given limited technical contributions, the upshot of the reported results are also weak in the sense that they are limited to text games. While the authors motivate this work by noting the potential transfer to real-world tasks from text, this work does not show empirical evidence of this transfer.

**Questions:**

1. Is it correct that prior results do not use gold trajectories while the proposed technique does? If so, this is a further limitation of the empirical results shown in this work. This would also open a variety of other sanity-check baselines. For instance, what is the performance of behavioral cloning without a decision transformer?
2. Figure 3 seems to show sample efficiency, however prior work are shown as flat lines. Why is this?
3. What is the statistical significance of this method compared to alternative methods? I am not sure how to interpret Table 2. For instance, sorcerer has a best score of 124 but a standard deviation of 90.6. What happens to this standard deviation when averaging in Figure 3? Are the performance gains statistically significant?

---

### Official Review · Reviewer_r7hd · 2023-11-01

**Soundness:** 2 fair
**Presentation:** 1 poor
**Contribution:** 1 poor
**Rating:** 3
**Confidence:** 4

**Summary:**

They present a method for training language models with in a goal conditioned way to solve text-game tasks. Their method involves training the LM to predict goal, action, next observation pairs. Unlike standard decision transformers, they sample the goal using a distribution exponentially tilted by the value of the goal. They demonstrate strong performance on text-game tasks and perform many ablations showing that their 1) their exponential tilt is very effective at choosing the right goal when \alpha is set high enough, 2) their next-observation auxiliary loss is necessary for maximizing performance, and 3) the choice of what goal representation to use can impact performance.

**Strengths:**

* Their method preforms very well empirically on these challenging text-game tasks.
* They abate the important design decisions behind their method
* They compare their method against many different baselines, showing improvements over all of them
* Their method is clean and simple

**Weaknesses:**

* The paper is not well written; the figures are low quality.
* The method has questionable novelty, as many of its ideas have been explored in some form in previous works. However, many of these works have not applied these specific ideas to text-games.
* Many of the baselines compared to are online methods, which likely perform worse than their offline method because of the exploration challenges presented by these long-horizon, sparse-reward text games. In the offline setting this challenge is alleviated because they are given at least 1 good trajectory and many partially good trajectories. This difference in difficulty is likely significant. It would be great if they could compare to some other offline methods, like filtered BC or [ILQL](https://arxiv.org/abs/2206.11871), trained on the same dataset.

Nitpicks:
* The statement “[offnline RL] is more difficult than online RL” is not necessarily true if your dataset is really good for example, then online RL will have many more challenges with exploration to catch up. Offline RL presents different challenges, but it is not fair to say that offline RL is strictly more difficult.
* "we convert distributions over discrete token representations of goal conditions into continuous ones": this statement was confusing to me because the distribution is still intently discrete after your exponential tilting.

**Questions:**

* Do the baseline results come from the original papers, or did you run them yourself? And if so, how did you tune the baselines?
* How does the method perform as you change the dataset quality? If you collect more low quality trajectories to mix in with the data, does it still perform well?

---

### Official Review · Reviewer_BF53 · 2023-11-11

**Soundness:** 2 fair
**Presentation:** 3 good
**Contribution:** 1 poor
**Rating:** 3
**Confidence:** 3

**Summary:**

This paper presents an approach to train a Reinforcement Learning (RL) agent leveraging language models to resolve Jericho text-based games. The training adopts the offline paradigm, where it samples trajectories by performing variation upon the max-human-score trajectory for each game. Two main adaptation is proposed during the process: 1) the goal scores are predicted (sampled with exponential tilting) in the learning sequence, and different strategies for goal conditioning are experimented; 2) an auxiliary task to predict next state is added. Empirical results on five Jericho games provide analysis on the impact of each proposed adaptation, and show that the offline models perform better than previous online RL approaches.

**Strengths:**

- This work shows an effective adaptation for offline RL training to resolve Jericho games, and the proposed model indeed outperforms several previous baselines.

- Certain insights regarding the impact of each proposed adaptation are provided by the empirical analysis.

**Weaknesses:**

I have some major concerns as follows regarding the technical significance and the evaluation:

- The adaptation proposed in this work is rather technically limited, as the approach adopts previous techniques directly (e.g. goal conditioning, next state prediction). The exponential tilting is also trivial to perform. Overall, this work focuses on the adaptation and its analysis, without bringing enough technical contributions.

- For the evaluation, there should be thorough comparison between the proposed offline training versus the online RL training. Since for the offline data generation, random variation is performed on the trajectory with the highest human scores, the distribution of the sampled trajectories may not reflect the real distribution as online exploration, thus it is not fair to simply compare online vs. offline by Figure 3, and the success of the offline model may rely more on the data generation, rather than the approach itself. There are not enough analysis provided comparing the generated offline data with online data.

**Questions:**

- Is it possible to show the offline training performance without using the trajectory of the highest human score?

- During inference, does the proposed offline-trained model show the potentials to outperform the upper-bound trajectory used in the training?